# Histological Transformation after Acquired Resistance to the Third-Generation EGFR-TKI in Patients with Advanced *EGFR*-Mutant Lung Adenocarcinoma

**DOI:** 10.3390/medicina58070908

**Published:** 2022-07-08

**Authors:** Po-Hsin Lee, Yen-Hsiang Huang, Ho Lin, Kuo-Hsuan Hsu, Kun-Chieh Chen, Jeng-Sen Tseng, Gee-Chen Chang, Tsung-Ying Yang

**Affiliations:** 1Division of Chest Medicine, Department of Internal Medicine, Taichung Veterans General Hospital, No.1650, Sect. 4, Taiwan Boulevard, Taichung 407, Taiwan; berry7bo@gmail.com (P.-H.L.); waynehuang0622@gmail.com (Y.-H.H.); jonyin@gmail.com (T.-Y.Y.); 2College of Medicine, National Yang Ming Chiao Tung University, No.155, Sec. 2, Linong St., Taipei 112, Taiwan; 3Program in Translational Medicine, National Chung Hsing University, No. 145, Xingda Rd., Taichung 402, Taiwan; 4Rong Hsing Research Center for Translational Medicine, National Chung Hsing University, No. 145, Xingda Rd., Taichung 402, Taiwan; 5Institute of Biomedical Sciences, National Chung Hsing University, No. 145, Xingda Rd., Taichung 402, Taiwan; geechen@gmail.com; 6Department of Life Sciences, National Chung Hsing University, No. 145, Xingda Rd., Taichung 402, Taiwan; hlin@dragon.nchu.edu.tw; 7Division of Critical Care and Respiratory Therapy, Department of Internal Medicine, Taichung Veterans General Hospital, No.1650, Sect. 4, Taiwan Boulevard, Taichung 407, Taiwan; vghryan@gmail.com; 8Division of Pulmonary Medicine, Department of Internal Medicine, Chung Shan Medical University Hospital, No.110, Sec. 1, Jianguo N. Road, Taichung 402, Taiwan; ckjohn@mail2000.com.tw; 9School of Medicine, Chung Shan Medical University, No.110, Sec. 1, Jianguo N. Road, Taichung 402, Taiwan; 10Institute of Medicine, Chung Shan Medical University, No.110, Sec. 1, Jianguo N. Road, Taichung 402, Taiwan; 11Department of Post-Baccalaureate Medicine, College of Medicine, National Chung Hsing University, No. 145, Xingda Rd., Taichung 402, Taiwan

**Keywords:** histological transformation, lung cancer, third-generation EGFR-TKI

## Abstract

*Background and Objectives*: Third-generation epidermal growth factor receptor (EGFR)-tyrosine kinase inhibitor (TKI) is one of the standard-of-care therapies in patients with *EGFR*-mutant lung adenocarcinoma; however, acquired resistance inevitably developed. Despite the proposition of histological transformation being one of the resistance mechanisms, its incidence and influence on outcome remain unclear. *Materials and Methods*: This was a retrospective study conducted at Taichung Veterans General Hospital on patients with advanced *EGFR*-mutant lung adenocarcinoma receiving the third-generation EGFR-TKI. Only patients receiving rebiopsy were included in the analysis. *Results*: A total of 55 patients were studied. Eight patients (14.5%) showed histological transformation, including three small cell carcinoma, three squamous cell carcinoma, one large cell neuroendocrine carcinoma, and one with a mixture of adenocarcinoma and squamous cell carcinoma components. The median treatment duration of the third-generation EGFR-TKI before rebiopsy was numerically longer in patients with histological transformation than those without (16.0 vs. 10.9 months). Both the overall survival time from the start of third-generation EGFR-TKI initiation (30.8 vs. 41.2 months) and from rebiopsy (6.6 vs. 12.9 months) to mortality were numerically shorter amongst the transformed population. All patients in the transformed group did not respond to the next line of systemic treatment. One patient with histological transformation receiving local treatment for the metastatic site had a longer overall survival. *Conclusions*: Repeating biopsy to identify histological transformation should be considered in patients with progression to the third-generation EGFR-TKI. Histological transformations could contribute to the acquired resistance with the implication of a worse prognosis. Further studies are needed to determine the optimal therapy for these patients.

## 1. Introduction

Epidermal growth factor receptor (EGFR)-tyrosine kinase inhibitors (TKIs) have been used, since their invention, as the first-line treatment of non-small-cell lung cancer (NSCLC) patients harboring *EGFR* mutations. The third-generation EGFR-TKI is one standard-of-care therapy for patients with *EGFR*-mutant lung adenocarcinoma [1]. However, TKI is not a curative treatment for lung cancer, with acquired resistance inevitably developed [2]. The resistance mechanisms to third generation covered both *EGFR*-dependent as well as independent ones. The possible molecular mechanisms of resistance include on-target mutations or amplified *EGFR* pathways, activation of those molecular pathways bypass or downstream, epithelial–mesenchymal transition, and histological transformation [3]. In lung adenocarcinoma patients experiencing disease progression to EGFR-TKI, histological transformation into small-cell lung cancer (SCLC) [4,5], squamous-cell carcinoma (SqCC) [6], and sarcomatoid carcinoma are known [7,8,9]. Concomitant mutations of *p53* and *Rb1* predict a higher risk of subsequent SCLC transformation [4].

Despite reports on histological transformation being one of the mechanisms of resistance to the third-generation EGFR-TKI, the incidence of histological transformation and its influence on patient outcomes remain unclear. Most research studies on such histological transformations focused on patients with acquired resistance to first- or second-generation EGFR-TKI and the incidence varies from 5% to 14% [10,11,12]. Our present study aimed to identify the incidence of histological transformation after the third-generation EGFR-TKI treatment and to characterize the clinical prognoses of these patients.

## 2. Material and Methods

### 2.1. Patients

This retrospective study was conducted on patients diagnosed with *EGFR*-mutant lung adenocarcinoma treated at the Taichung Veterans General Hospital in Taiwan. All patients received the third-generation EGFR-TKI during the period from January 2014 to June 2021. We enrolled patients who received rebiopsy after the disease progression to the third-generation EGFR-TKI. Those patients who had received rebiopsy within 3 months after the start of the third-generation EGFR-TKI were excluded as they indicated primary resistance to the treatment. Patients with wild-type or unknown *EGFR* mutation status were also excluded. Our study was approved by the Institutional Review Board of Taichung Veterans General Hospital (IRB No. CF12019 and No. CF15271). Written informed consents for clinical data records and genetic testing were obtained from all patients. Experiments were carried out in accordance with the approved guidelines and regulations.

### 2.2. Data Records and Response Evaluation

Clinical data for our analysis included patients’ age, gender, smoking status, the Eastern Cooperative Oncology Group performance status (ECOG PS), histological types, *EGFR* mutation status, tumor stage, history of treatments, and survival status. The staging of lung cancer TNM (tumor, node, and metastases) was conducted according to the American Joint Committee on Cancer (AJCC) staging system (8th edition) [13]. Unidimensional measurements as defined by Response Evaluation Criteria in Solid Tumors (RECIST) version 1.1 were adopted [14]. Oncogenic mutation analyses of *EGFR* status were performed according to the manufacturer’s protocol for the MassARRAY system (Sequenom, San Diego, CA, USA) (details see previously published report) [15]. Some patients received analysis by the cobas *EGFR* mutation test v2 (Roche MolecularSystems, Inc, Branchburg, NJ, USA).

### 2.3. Statistics Analyses

To compare intergroup differences for categorical and continuous variables, Fisher’s exact test and Mann–Whitney U test were used, respectively. Two-tailed *p* values  <  0.05 were considered statistically significant differences. The overall survival (OS) was estimated using the Kaplan–Meier method, whereas the between-group differences were assessed using the stratified log-rank test. OS was analyzed from the start of the third-generation EGFR-TKI treatment to mortality and from rebiopsy to mortality. All analyses were performed with the IBM SPSS Statistics package, version 23 (IBM Corporation, Armonk, NY, USA).

## 3. Results

### 3.1. Patient Characteristics

The selection algorithm of participants is illustrated in Figure 1, and patient characteristics are shown in Table 1. A total of 481 patients received the third-generation EGFR-TKI within the study period. Rebiopsy was made after disease progression to the third-generation EGFR-TKI in 60 patients. A total of 55 patients were analyzed, after excluding five patients (three receiving rebiopsy within three months of EGFR-TKI initiation, one with wild-type *EGFR*, and one with unknown *EGFR* status).

Their median age was 58 years. Of these patients, 34 (61.8%) were female, 44 (80%) were non-smokers, and eight (14.5%) received the third-generation EGFR-TKI as first-line treatment. Regarding the types of third-generation EGFR-TKI, most patients (*n* = 53) received osimertinib, while the other two received almonertinib. Baseline *EGFR* mutations of these patients are as follows: 22 (40%) *EGFR* exon 19 deletion (19Del), 19 (34.5%) exon 21 L858R substitution, nine (16.4%) primary exon 20 T790M, and five (9.1%) uncommon or complex mutation. In addition to those patients with primary T790M, another 30 (54.5%) patients had acquired T790M before the third-generation EGFR-TKI treatment. The median duration of the third-generation EGFR-TKI treatment before rebiopsy was 11.0 months. The median duration from rebiopsy to third-generation EGFR-TKI discontinuation was 1.0 month.

### 3.2. Comparison of Characteristics of Patients with/without Histological Transformation

The characteristics of patients with or without histological transformation after the third-generation EGFR-TKI treatment are shown in Table 1. The histological transformation was found in eight (14.5%) patients, while the pathologic results of the remaining 47 (85.5%) patients remained adenocarcinoma. No patients receiving the third-generation EGFR-TKI as first-line treatment had developed the histological transformation. More males had histological transformation (five of 21 patients (23.8%) vs. three of 34 patients (8.8%)). The median treatment time of the third-generation EGFR-TKI before rebiopsy was longer in patients with histological transformation than in those without (16.0 vs. 10.9 months). Such differences were statistically insignificant.

### 3.3. Clinical Outcomes and Overall Survival in Patients with Histological Transformation

Clinical data and outcomes of patients with histological transformation are shown in Table 2. In terms of transformed histological types, three patients were squamous cell carcinoma (SqCC), three were small cell lung carcinoma (SCLC), one was large cell neuroendocrine carcinoma (LCNEC), and one was a mixture of adenocarcinoma and squamous cell carcinoma components. One patient experienced two transformations, initially SqCC and later sarcomatoid carcinoma [7].

Three patients received genetic testing from tissues obtained from the rebiopsy. The initial *EGFR* status of patient No. 1 was exon 21 L858R. She developed *EGFR* T790M after resistance to erlotinib and chemotherapy. The *EGFR* status maintained L858R plus T790M in the rebiopsy specimen. In patient No.2, the initial *EGFR* mutation assay showed 19Del. After showing resistance to gefitinib and chemotherapy, T790M developed. He experienced squamous cell carcinoma transformation, and the *EGFR* status revealed 19Del plus T790M. Another episode of histological transformation into sarcomatoid carcinoma occurred 11 months later and the *EGFR* mutation assay showed 19Del as well as loss of T790M mutation. Patient No. 8 had received an operation for lung cancer, but the tumor recurred. He harbored *EGFR* L858R mutation and gefitinib was prescribed. He underwent the third-generation EGFR-TKI treatment after disease progression, despite negative T790M findings in the rebiopsy specimen. Small cell transformation developed 5.1 months later and the next-generation sequencing using the FoundationOne CDx method was performed. Results were mutations of *EGFR* L858R, *RB1* p.R445X, and *TP53* c.96 + 1G > T.

Aside from two patients who deteriorated and died shortly after histological transformation, the remaining six patients in the transformation group did not respond to the next line of treatment. Patients No. 5 and No. 6 had already received the etoposide before histological transformation, so we did not prescribe platinum plus etoposide as the next line of treatment for the histological change. Patient No. 4 already received platinum and gemcitabine before histological transformation, so docetaxel and etoposide were prescribed after squamous cell transformation.

Both overall survivals from the start of third-generation EGFR-TKI treatment (30.8 (95% CI, 21.6–40.0) vs. 41.2 (95% CI, 24.5–57.9) months, *p* = 0.174) and from the start of rebiopsy (6.6 (95% CI, 3.5–9.7) vs. 12.9 (95% CI, 10.6–15.3) months, *p* = 0.100) were numerically shorter amongst the transformed population (Figure 2A,B). There was no statistical difference in survival between patients with SqCC transformation and with SCLC transformation. Overall survivals from the start of third-generation EGFR-TKI treatment were 30.8 (95% CI, 3.4–58.2) for patients transformed into SqCC, and 23.8 (95% CI, 21.8–25.8) months for patients transformed into neuroendocrine carcinoma (including SCLC and LCNEC); whereas survivals from the start of rebiopsy were 2.5 months (95% CI, 0.0–10.9) and 6.6 months (95% CI, 4.2–9.0), respectively. We noted the limited number of patients developing histological transformation had wide-ranged survival times.

## 4. Discussion

The reported incidence of histological transformation from NSCLC to SCLC in patients treated with first- or second-generation EGFR-TKI varies from 5% to 14% [10,11,12]. In the study conducted by Sequist et al., 8 (21.6%) of the 37 NSCLC patients harboring *EGFR* mutations who had tumor biopsy after acquiring drug resistance to first- or second-generation EGFR-TKI showed phenotypic changes. Amongst those with histological transformation, five (14%) were SCLC transformation, two were epithelial-to-mesenchymal transition (EMT), and the remaining one was sarcomatoid transformation. The median time from TKI start to small cell transformation was 22.0 months (range 14.0 to ≥36.0 months), whereas the median time to histological transformation (SCLC, EMT, and sarcomatoid transformation) was 16.0 months (range 11.0 to ≥36.0 months) [12]. In another study, which analyzed 58 *EGFR*-mutant NSCLC patients developing small cell transformation after EGFR-TKI treatment, their median time since diagnosis of advanced NSCLC to SCLC transformation was 17.8 months; however, the EGFR-TKI prescribed in the study was not limited to the first- or second-generation, and 33% of these patients received the third-generation EGFR-TKI treatment. The median overall survival since diagnosis was 31.5 months, whereas the median survival since the time of SCLC transformation was only 10.9 months [5]. In our present study, we shared clinical themes similar to the previous studies. In total, 4 out of our 55 patients transformed into neuroendocrine carcinoma (7.3%). The median time from TKI start to histological transformation was 16.0 months. The overall survival of patients transforming into SCLC from the start of the third-generation EGFR-TKI was 23.8 months, and from rebiopsy was 6.6 months.

The actual incidence of SqCC transformation after first- or second-generation EGFR-TKI treatment remains unknown. Clery et al. reviewed 15 lung adenocarcinoma patients with activated *EGFR* mutations. They developed the SqCC transformation after receiving first- or second-generation EGFR-TKIs. SqCC transformation mostly occurred in female patients (86.7%). Apart from the original *EGFR* mutation, 2 of the 15 cases showed an additional *PIK3CA* mutation in the rebiopsy samples [16]. Another case series regarding NSCLC patients with changed phenotype to SqCC after EGFR-TKI treatment showed an 11.5 month median time from TKI start to SqCC transformation. Overall survival from NSCLC diagnosis was 20 months or from SqCC onset was 3.5 months [6]. In our present study, the rate of SqCC transformation after the third-generation EGFR-TKI was 7.3%. The median time from TKI start to SqCC transformation was 13.3 months. Overall survival of patients transforming into SqCC from the start of third-generation EGFR-TKI was 30.8 months and from rebiopsy was 2.5 months.

Regarding histological transformation after third-generation EGFR-TKI treatment, Schoenfeld et al. analyzed post-osimertinib-treated tumor tissues from 71 patients. Histological transformation after osimertinib treatment occurred in three of 16 (18.8%) patients with osimertinib as first-line treatment and in seven of 55 (12.7%) patients with osimertinib as second-line or later treatment. Their overall transformation rate was therefore 14.1%. Amongst these patients, their transformed histological types were SqCC (five patients) and SCLC (five patients) [17]. In our present study, the overall transformation rate was similar (14.5%). The rate of SqCC transformation was actually identical to neuroendocrine carcinoma transformation (7.3% vs. 7.3%). None of our patients using the third-generation EGFR-TKI as first-line treatment had developed any histological transformation; however, only eight patients using first-line third-generation EGFR-TKI had received rebiopsy in our cohort.

In previous studies, both platinum plus etoposide and taxanes have yielded high response rates after histological transformation [5,18]. Ferrer et al. analyzed 48 patients with *EGFR*-mutant NSCLC and 13 with non–*EGFR*-mutant who had developed small cell transformation and 45% of the patients with *EGFR*-mutant and 40% of non-*EFGR*-mutant patients achieved an objective response to platinum plus etoposide treatment [18]; however, Ho et al. put forward a different opinion. They found that *Sin3* and *NuRD* pathways, which are two major histone deacetylase (HDAC) complexes, were up-regulated in the *EGFR*-mutant SCLC. Furthermore, trichostatin A, a pan-HDAC inhibitor, has better inhibitory efficacy than cisplatin-etoposide in SCLC-transforming cells [19]. In our study, those patients who transformed into SCLC did not respond to platinum plus etoposide treatment. Moreover, the next-line treatment for patients with histological transformation into SqCC was also ineffective. Most of our patients had dismal outcomes after histological transformation. One exception is patient No. 8. He had disease progression into pleural metastasis after osimertinib treatment. He received an excisional biopsy of the metastatic tumor and the pathologic finding was SCLC. Cisplatin, etoposide, and atezolizumab were therefore prescribed; however, bilateral lower limb weakness appeared and an MRI of the T-L spine showed T spine vertebral metastasis with intra-dural invasion. Laminectomy with tumor removal was performed and the pathologic findings remained SCLC. Subsequent radiotherapy for the metastatic sites was applied. He was further treated with paclitaxel, bevacizumab, and atezolizumab. He remained in relatively stable conditions until now (no evidence of disease progression for ≥7 months). Yu et al. proposed that local therapy is an optional treatment strategy for patients with advanced *EGFR*-mutant lung cancers and with acquired resistance to EGFR-TKI [20]. Further research works are required to explore appropriate treatments and to evaluate the role of local ablative treatment for patients with such histological transformation.

The mechanisms of histological transformation are under investigation. Marcoux et al. studied 67 patients with *EGFR*-mutant NSCLC undergoing SCLC transformations. In addition to the original *EGFR* mutation, the more common mutations in SCLC samples include *TP53* (79%) and *Rb1* (58%) [5]. In another study, 5% of patients with *EGFR*-mutant lung cancers showed concurrent *RB1* and *TP53* changes. These patients typically have a high risk for small cell transformation (18%) [21]. Aside from *RB1* and *TP53* losses, the *SOX* family mutation, *PI3K/AKT* pathway, *MYC* and *AURKA* amplification, as well as *Notch* signaling downregulation, may underlie mechanisms of transformation [22]. In regard to squamous cell transformation, Schoenfeld et al. characterized the mechanisms of resistance to osimertinib. Five patients with squamous transformation presented with considerable genomic complexity, including acquired the *PIK3CA* mutation, chromosome 3q amplification, and *FGF* amplification [23]. Furthermore, patients with a transformation from adenocarcinoma to sarcomatoid carcinoma after EGFR-TKI treatment experience epithelial-to-mesenchymal transitions [24]. Tumors sometimes metastasize to unusual sites such as the pancreas, kidney, and digestive tract in patients with sarcomatoid transformation [25]. These patients were found to have *MET* amplification, *MET* copy number gains, and high levels of PD-L1 expression [8,9].

The major limitations of this study are as follows. First, its retrospective nature and the limited number of cases receiving rebiopsy after third-generation EGFR-TKI. Second, rebiopsy after progression to third-generation EGFR-TKI was not performed for all patients. Third, limited by the available biopsy specimens, we were unable to clearly determine whether the histological types other than adenocarcinoma had come from a post-EGFR-TKI treated transformation or from pre-existed mixed histology. Fourth, not all the patients with histological transformation received biopsy before the initiation of third-generation EGFR-TKI; therefore, the timing of histological transformation development was not clear. Because of these limitations, our results should be interpreted with caution.

## 5. Conclusions

Rebiopsy should be considered in patients with progression to the third-generation EGFR-TKI to identify histological transformation. The possible histological types of transformation include SCLC, SqCC, and sarcomatoid carcinoma. Histological transformations may contribute to the acquired resistance and may imply a worse prognosis. Local therapy is likely an option for patients not well responding to systemic antineoplastic treatments.

## Figures and Tables

**Figure 1 medicina-58-00908-f001:**
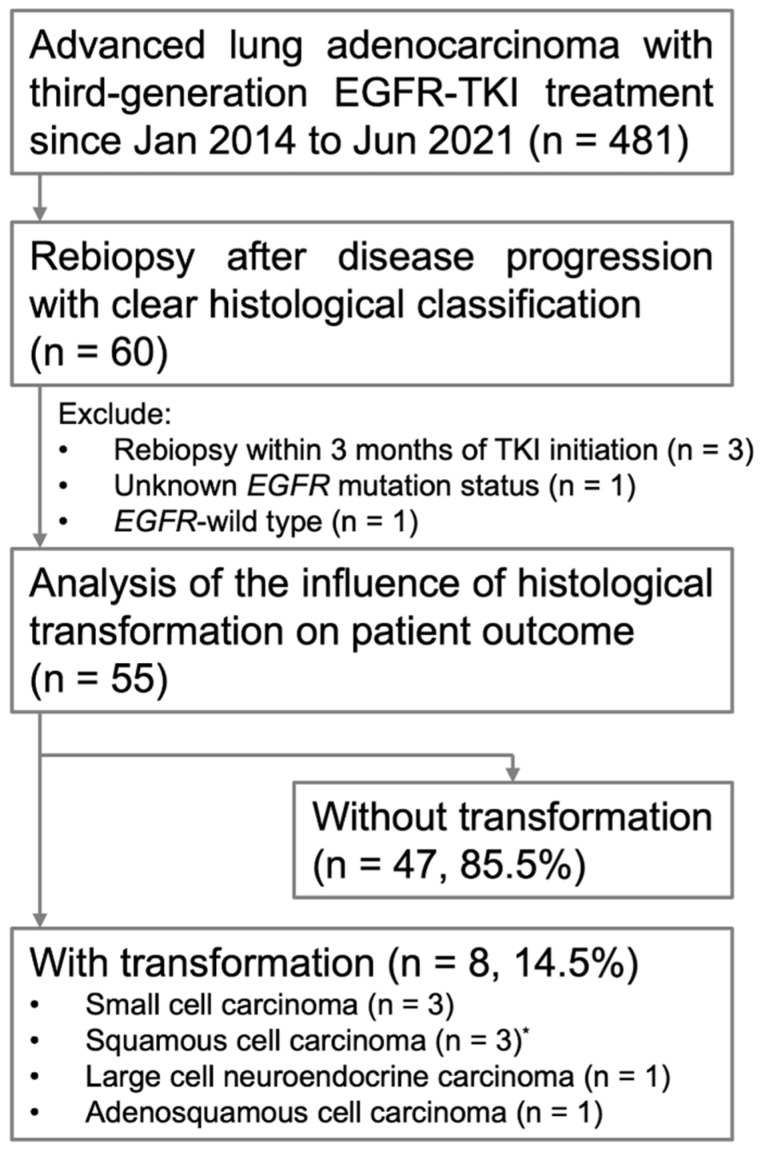
Algorithm for inclusion of study participants. * Includes one patient with transformation into squamous cell carcinoma in the first rebiopsy and into sarcomatoid carcinoma in the second rebiopsy.

**Figure 2 medicina-58-00908-f002:**
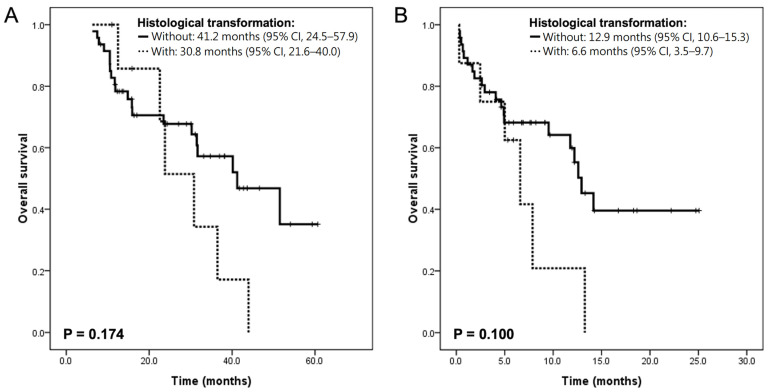
Influence of histological transformation on overall survey. Survival from the start of the third-generation EGFR-TKI to mortality (**A**) or from rebiopsy to mortality (**B**).

**Table 1 medicina-58-00908-t001:** Characteristics of patients with or without histological transformation after the third-generation EGFR-TKI treatment.

	Total (N = 55)	Without Transformation (N = 47)	With Transformation (N = 8)	*p* Value ^&^
Age, median (IQR; 25–75%)	58 (53.0–62.6)	59.0 (53.0–65.0)	55.0 (51.5–59.0)	0.262
Gender, *n* (%)				0.238
Male	21 (38.2%)	16 (34%)	5 (62.5%)	
Female	34 (61.8%)	31 (66%)	3 (37.5%)	
Smoking, *n* (%)				0.654
No	44 (80.0%)	38 (80.9%)	6 (75%)	
Yes	11 (20.0%)	9 (19.1%)	2 (25%)	
ECOG PS, *n* (%)				1.000
0–1	53 (96.4%)	45 (95.7%)	8 (100%)	
≥2	2 (3.6%)	2 (4.3%)	0 (0%)	
Third-generation EGFR-TKI as first-line treatment, *n* (%)				0.587
Yes	8 (14.5%)	8 (17%)	0 (0%)	
No	47 (85.5%)	39 (83%)	8 (100%)	
Baseline *EGFR* mutation, *n* (%)				0.754
19del	22 (40.0%)	18 (38.3%)	4 (50.0%)	
L858R	19 (34.5%)	16 (34.0%)	3 (37.5%)	
Uncommon or compound mutation	5 (9.1%)	5 (10.6%)	0 (0%)	
Primary T790M	9 (16.4%)	8 (17.0%)	1 (12.5%)	
19del + T790M	5 (9.1%)	4 (8.5%)	1 (12.5%)	
L858R + T790M	3 (5.5%)	3 (6.4%)	0 (0%)	
19del + G719C + T790M	1 (1.8%)	1 (2.1%)	0 (0%)	
Emergence of T790M, *n* (%)				0.520
Yes	30 (54.5%)	25 (53.%)	5 (62.5%)	
No	8 (14.5%)	6 (12.8%)	2 (25.0%)	
N/A	8 (14.5%)	8 (17.0%)	0 (0%)	
Primary T790M	9 (16.4%)	8 (17.0%)	1 (12.5%)	
Treatment duration of third-generation EGFR-TKI before biopsy(months), median (IQR; 25–75%)	11.0 (6.0~19.1)	10.9 (5.7~21.9)	16.0 (7.9~18.1)	0.489
Interval between third-generation EGFR-TKI discontinuation to rebiopsy(months), median (IQR; 25–75%)	−1.0 (−5.1~2.2)	−1.0 (−5.1~2.2)	−1.1 (−5.1~2.7)	0.842

ECOG PS, Eastern Cooperative Oncology Group Performance Status; *EGFR*, epidermal growth factor receptor; TKI, tyrosine, kinase inhibitor; T790M, Thr790Met; N/A, not available. **^&^** Age, Treatment duration of third-generation EGFR-TKI, Interval between third-generation EGFR-TKI discontinuation to rebiopsy by Mann–Whitney U test; otherwise by Fisher’s exact test.

**Table 2 medicina-58-00908-t002:** Clinical data and outcomes of lung cancer patients developing histological transformation after the third-generation EGFR-TKI treatment. F, female; M, male; L858R, Leu858Arg; 19del; exon 19 deletion; T790M, Thr790Met; C/T, chemotherapy; R/T, radiotherapy; OP, operation; ADC, adenocarcinoma; SqCC, squamous cell carcinoma; LCNEC, large cell neuroendocrine carcinoma; SCLC, small cell carcinoma; N/A, not available; PD, disease progression.

No	1	2	3	4	5	6	7	8
Age, years	55	44	53	62	56	51	55	60
Gender	F	M	F	M	M	M	F	M
Smoking	N	N	N	Y	N	Y	N	N
Baseline *EGFR* mutation	L858R	19del	19del + T790M	19del	L858R	19del	19del	L858R
Treatment before third-generation EGFR-TKI	C/T, erlotinib	gefitinib, C/T	afatinib	afatinib, C/T, erlotinib	erlotinib, R/T, C/T	erlotinib, C/T	gefitinib, C/T	OP, gefitinib
Pathologic finding of biopsy before third-generation EGFR-TKI	Pericadial effusion cell block: ADC	Primary tumor rebiopsy: ADC	Primary tumor rebiopsy: ADC	Metastatic lung tumor biopsy: ADC	Primary tumor rebiopsy: ADC	Primary tumor rebiopsy: ADC	Primary tumor rebiopsy: ADC	Biopsy not performed
Emergence of T790M before third-generation EGFR-TKI	Y	Y	Primary T790M	N	Y	Y	Y	N
Third-generation EGFR-TKI treatment duration, months	10	42.2	16.5	7.2	15.9	16	18.6	5.1
Rebiopsy site	Tongue	Spine (1st), chest wall (2nd)	Bone	Lung	Lung	Liver	Lung	Pleura
Histological transformation	SqCC	SqCC (1st), sarcomatoid (2nd)	mixture of ADC and SqCC	SqCC	LCNEC	SCLC	SCLC	SCLC
Genetic test from rebiopsy tissue	L858R + T790M	19del + T790M(1st), 19del alone(2nd)	N/A	N/A	N/A	N/A	N/A	L858R, *RB1*, *TP53* (by NGS)
Treatment after transformation	gemcitabine + R/T	OP + R/T + gemcitabine	hospice	docetaxel + etoposide	irinotecan	carboplatin + irinotecan	carboplatin + etoposide	carboplatin + etoposide
Response to the treatment	N/A	PD	N/A	PD	PD	PD	PD	PD

## Data Availability

The data presented in this study are available on request from the corresponding author.

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
