# Peer review of "Histological Transformation after Acquired Resistance to the Third-Generation EGFR-TKI in Patients with Advanced EGFR-Mutant Lung Adenocarcinoma"

_medicina, 2022, doi:10.3390/medicina58070908_

Round 1

Reviewer 1 Report

Lee et al did a research which focused on pathological transformation of patients getting resistance to the third generation TKIs. They analyzed characteristics of these special patients and compared their outcome with those who did not having pathological transformation. And they indicated that patients having pathological transformation might have a worse outcome. It is an interesting study but there are some small concerns should be further addressed.

1.       It would be better to clarify the exact third generation EGFR-TKI that used for included patients.

2.       As authors showed that all patients having resistance to the third generation EGFR-TKI due to pathological transformation had went through multiple lines treatment, whether a relatively worse overall survival should not just be blamed on pathological transformation. How many lines of therapy that patients had had should also be considered as a confounder. Cox regression analysis might be a way to analyzed whether pathological transformation was an independent factor for worse outcome of patients.

3.       Table 2 was not shown completely as information of patients 7 and 8 were out of the bound of paper.

4.       It would be better to write EGFR in italic when it was interpreted as a gene. Some in abstract were not italic.

5.       Some gramma mistakes occurred. For example, a double “at” in this sentence “This was a retrospective study conducted at at Taichung Veteran General Hospital …”and Our present study was aimed to identify…” should be “Our present study aimed to identify…”. The gramma should be checked again.

Reviewer 2 Report

The authors demostrate histological transformation (HT) after third-generation EGFR-TKI treatment for EGFR-mutant adenocarcinoma patients. Histological transformation is one of major resistant mechanisms against EGFR-TKI, especially third-generation TKI (osimertinib). The authors show the proportion of HT after osimertinib treatment who underwent re-biopsy, and detailed clinical information of them are provided. The theme of the article excites interest, however, there are some points to be clarified.

Major comments:

1. HT as resistant mechanism is one of the characteristic features for third-generation EGFR-TKI, but it is also known to be observed after treatment with 1st- and 2nd-generation TKIs such as gefitinib and erlotinib. The reported cases of HT in this article seem to have received 1st- and 2nd-generation TKIs in addition to osimertinib between diagnosis and re-biopsy. Therefore, association between HT and 3rd-generation TKI is not clear from the presented cases as the authors mention in limitation paragraph. I recommend such aspect should be clearly noted and whether biopsy specimens were available just before osimertinib treatment should be included in Table 2.

Minor comment:

1. One "adenosquamous cell carcinoma" transformation after osimertinib treatment is reported (Table 2, etc). The terminology is not correct (adenosquamous carcinoma is correct), and the term should be used for resected specimen in which we can examine whole of the tumor according to WHO Classification of Tumours, because adenosquamous carcinoma is defined as tumor made up of adenocarcinoma and squamous cell carcinoma component and both of them must comprise >10% of entire tumor. I am not sure how the histology of re-biopsy specimen of the patient was like, but I recommend provide the detail of the histology, not using the "adenosquamous carcinoma" term (e.g. "mixture of adenocarcinoma and squamous cell carcinoma component").

Round 2

Reviewer 1 Report

None.